# Genomic Changes Driven by Radiation-Induced DNA Damage and Microgravity in Human Cells

**DOI:** 10.3390/ijms221910507

**Published:** 2021-09-29

**Authors:** Afshin Beheshti, J. Tyson McDonald, Megumi Hada, Akihisa Takahashi, Christopher E. Mason, Maddalena Mognato

**Affiliations:** 1KBR, NASA Ames Research Center, Space Biosciences Division, Moffett Field, CA 94035, USA; 2Stanley Center for Psychiatric Research, Broad Institute of MIT and Harvard, Cambridge, MA 02142, USA; 3Department of Radiation Medicine, Georgetown University School of Medicine, Washington, DC 20007, USA; jm3376@georgetown.edu; 4Radiation Institute for Science & Engineering, Prairie View A&M University, Prairie View, TX 77446, USA; mehada@pvamu.edu; 5Gunma University Heavy Ion Medical Center, 3-39-22 Showa-Machi, Maebashi 371-8511, Gunma, Japan; a-takahashi@gunma-u.ac.jp; 6Department of Physiology and Biophysics, Weill Cornell Medicine, New York, NY 10065, USA; chm2042@med.cornell.edu; 7The World Quant Initiative for Quantitative Prediction, Weill Cornell Medicine, New York, NY 10065, USA; 8Department of Biology, University of Padova, Via U. Bassi 58/B, 35131 Padova, Italy

**Keywords:** space radiation, microgravity, OsaD, genetic and epigenetic changes, radiation carcinogenesis risk

## Abstract

The space environment consists of a complex mixture of different types of ionizing radiation and altered gravity that represents a threat to humans during space missions. In particular, individual radiation sensitivity is strictly related to the risk of space radiation carcinogenesis. Therefore, in view of future missions to the Moon and Mars, there is an urgent need to estimate as accurately as possible the individual risk from space exposure to improve the safety of space exploration. In this review, we survey the combined effects from the two main physical components of the space environment, ionizing radiation and microgravity, to alter the genetics and epigenetics of human cells, considering both real and simulated space conditions. Data collected from studies on human cells are discussed for their potential use to estimate individual radiation carcinogenesis risk from space exposure.

## 1. Introduction

The space environment represents a substantial hazard for human transit and permanent settlement outside the protective magnetosphere. In the last 60 years of human spaceflight, cellular and tissue alterations have been reported because of human exposure to the space environment. The first evidence of human body alterations comes from epidemiological data collected from astronauts showing muscle weakness, loss of body mass, bone deterioration, cardiovascular disease, and immune suppression [1]. Further data have been collected from more recent space missions [2,3,4] and are visible on the National Aeronautics and Space Administration (NASA) website (https://humanresearchroadmap.nasa.gov/, accessed on 23 September 2021). NASA’s Human Research Program (HRP) has classified hazards for astronauts into five categories: space radiation, isolation, distance from Earth, gravity fields, and hostile/closed environments. Since all of these hazards are present together during a space journey, the risk due to space exposure is complicated to define. According to NASA’s HRP, radiation carcinogenesis risk assessment is considered a research priority for future space missions that aims at long term establishment of humans on Mars and the Moon. Radiation carcinogenesis derived from spaceflight represents a “big” risk that needs to be assessed before starting deep space missions. Such a risk includes cancer occurrence that would manifest in the subsequent years after a space mission. At present, the level of risk acceptance is set by NASA at 3% [5]. However, several individual genetic factors may predispose astronauts to a higher risk of radiation-induced reactive oxygen species (ROS) formation and DNA damage beyond the accepted level of radiation risk. Factors of individual-radiation sensitivity include genotype, age, diet, diabetes, etc. [6,7]. On the other hand, studies about the effects of space conditions on human molecular pathways are still limited as are data about space radiation effects on the human genetic and epigenetic landscape. Genomic alterations in the form of chromosome damage were found in peripheral blood lymphocytes (PBLs) from astronauts on 3–6 month-long missions [8]. However, data from long-duration missions are limited resulting in a limited number of studies on genetic alterations in space flown human cells. Among these, the NASA Twins Study was carried out in two monozygotic twin astronauts and reported gene expression changes in the astronaut that spent a 340 day-mission onboard the International Space Station compared with his Earth-bound twin [3]. Recently, Malkani et al. [9] identified a spaceflight associated miRNA signature in mammalian cells. This underlies the great potential of genomic studies in evaluating the cellular response to a space environment. Nevertheless, space risk prediction deals with many uncertainties, among which are primarily due to cosmic radiation and altered gravity fields. Taking into account these two parameters, here we summarize the knowledge on genetic and epigenetic pathway modulation in human cells exposed to real or simulated space conditions.

## 2. Space Radiation Environment

The space radiation environment is one of the major risk factors in long-term human exploration. In Low-Earth Orbit (LEO), such as for the International Space Station (ISS) with an altitude of approximately 400 km and an orbital inclination of 51.6°, astronauts and cosmonauts are exposed to a variety of radiation sources. This includes high-energy radiation from protons to high-atomic number iron nuclei (peak energies of 0.1–1 GeV/n) termed galactic cosmic rays (GCR), low-energy (mostly 50 MeV/n) protons from solar energetic particles (SEP), and medium-energy (<250 MeV/n) protons trapped in the Van Allen Belts [10,11]. Roughly half of the dose in LEO is expected to come from GCRs and trapped protons; all these sources are affected by the Earth’s magnetic field. In addition, these particles combine to produce a complex radiation environment in and around a LEO. The complexity of these radiations is dependent on orbital parameters (orbital inclination and altitude), the solar cycle, and shielding of the spacecraft. The dose rates inside of the ISS are about 0.3 to 0.4 mSv/day at the solar maximum and minimum, respectively [12]. These dose rates are about a few hundred times higher than that at sea level on Earth. Beyond the Earth’s magnetic field, high-energy charged particles of SEPs and GCRs strongly affect the dosimetry of astronauts and cosmonauts. The cosmic-ray dose rates during interplanetary travel, on the lunar surface, or on the Martian surface are about 0.4, 0.3, or 0.2 mSv/day at the solar maximum, respectively, and 1.1, 0.9, or 0.5 mSv/day at the solar minimum, respectively [12]. Although higher solar activity significantly reduces the dose from cosmic rays, the potentially large risk of a solar particle event (SPE) becomes higher during the solar maximum period.

Although the effects of GCRs are greatly concerning, space-based studies to investigate the health effects of space radiation have severe limitations. Ground-based studies have an important role in order to obtain statistically significant data. High-energy ion beams at ground-based accelerators such as the BEVALAC at Lawrence Berkeley National Laboratory (LBNL), the Helmholtz Center for Heavy Ion Research (GSI), NASA’s Space Radiation Laboratory (NSRL) at Brookhaven National Laboratory (BNL), and the Heavy Ion Medical Accelerator (HIMAC) at the National Institute of Radiological Sciences (NIRS) have been used to perform extensive research with a single beam or beam combinations. Recently, the NSRL has developed SPE and GCR simulated beams by using a rapid switching technology for various ion species and energies [13]. In addition, neutrons from a panoramic Californium-252 source in a concrete-shielded building on the campus of Colorado State University [14] offers a low-dose-rate exposure (1 mGy/day) to animals. Long-term exposure to neutrons is a potential health hazard when astronauts encounter GCRs during their missions outside the Earth’s magnetosphere. While neutrons constitute a small proportion of GCRs, secondary neutrons ejected owing to the interaction between GCRs and shielding elements are significant [15,16].

High-LET radiation effects on cells and animals have been studied extensively in ground-based facilities, however these conditions lack microgravity which is another major environmental stressor in space. In order to understand the effects of space radiation combined with microgravity conditions, a unique system has been developed by Takahashi’s group [17,18,19]. With these systems, samples can be exposed to X-rays, Carbon-ions or neutrons simultaneously with simulated microgravity.

## 3. Oxidative Stress and Damage (OsaD) Generation in the Space Environment

Cell exposure to ionizing radiation (IR) elicits a complex response which is driven by many proteins from different biological pathways that cooperate to defend cells from the cytotoxic and genotoxic insult. The biological response of IR is strictly related to physical variables of radiation, such as energy, LET, dose-rate, final dose, exposure-time [20], and also to the cellular context (cell type and proliferation status) [21]. The mixed nature of space radiation causes different forms of cellular damage, related to both the direct and indirect effects of IR. GCRs are comprised of high-energy protons as well as high charge (Z) and energy (E) nuclei (HZE) [5] that have a high ionizing power and cause severe damage to DNA molecules. When IR hits the DNA molecule, single-strand breaks (SSBs) and double-strand breaks (DSBs) arise. Once radiation passes throughout the cell and generates DSBs, the upstream signaling protein kinase ataxia telangiectasia mutated (ATM) is activated to trigger chromatin remodeling and a cascade of protein phosphorylation called the DNA-Damage Response (DDR). The ultimate goal of the DDR pathway is to overcome DNA lesions assuring cell survival and maintaining genome stability. This is hard work considering the dangerous power of space radiation. The DDR pathway relies on the activity of many different proteins that, once phosphorylated by ATM, take part in sensing/transducing the DNA damage signal to effector proteins in control of the DNA-repair machinery, cell cycle checkpoints (if cells are proliferating), and cell death by apoptosis [22,23,24]. In addition, ATM mediates DNA damage-induced changes in RNA metabolic pathways, including mRNA synthesis [25] and miRNA biogenesis [26]. DNA repair is activated by proteins of the PARP family that bind to DNA SSBs and DSBs and catalyze the ADP-ribosylation of chromatin proteins [22]. However, being that cells are mainly composed of water, many reactive oxygen species (ROS) are formed due to water radiolysis. This gives rise to high levels of free radicals (such as hydrogen peroxide, hydroxyl radical, and superoxide anion) that cause oxidative damage to many organic molecules including lipids, proteins, and nucleic acids. The same cellular targets are also damaged by highly reactive nitrogen species (RNS) that are a consequence of the early activation of nitric oxide synthases in irradiated cells [27]. The extent of oxidative DNA damage depends on the yield of ROS and RNS and is related to the quality, dose, and dose-rate of radiation.

The radiation-induced increase in oxygen and nitrogen reactive species alters the physiological equilibrium between reduction and oxidation giving rise to a condition called “Oxidative stress and Damage” (OsaD) [28] (Figure 1). The presence of OsaD induces a highly regulated defense response that on the one hand activates antioxidant enzymes (i.e., superoxide dismutase, catalase, and glutathione s-transferase) and low molecular weight antioxidants [29] while it cooperates with the DDR pathway. Indeed, ATM activation occurs not only in response to direct radiation-induced DNA damage, but also in response to indirect oxidative DNA damage [30], although only few proteins are shared by the OsaD and DDR pathways [31,32].

Oxidative DNA damage consists mainly of oxidized bases, among which 8-Oxo-7,8-dihydro-2′-deoxyguanosine (8-oxodGuo) is the most frequent and is used as a cellular biomarker of oxidative stress [33]. In addition to purines, pyrimidines are also damaged by ROS, leading to oxidized pyrimidine derivatives such as thymine glycol (Tg) and 5,6 dihydrouracil (DHU) that can block DNA and RNA polymerases [33]. The presence of ROS also creates oxidized base-derived apurinic/apyrimidinic sites and SSBs [34], whereas RNS mainly causes protein modifications in the form of tyrosine nitration and S-nitrosylation of cysteine that alter the normal activity of proteins [29]. Reactive oxygen and nitrogen species take part in intracellular signaling cascades regulating several physiological functions [35]. In this regard, several transcription factors are sensitive to changes in ROS/RNS levels and can activate specific signaling pathways [29]. The amount of direct and indirect DNA damage is strictly related to the radiation quality and dose but may also be observed in cells that were not directly irradiated giving rise to a non-targeted (bystander) effect [27,36]. Not only DNA but also ribosomal RNA undergoes oxidative damage in the presence of oxidative stress [37]. Notably, mitochondria, by generating mitochondrial ROS during cellular respiration, contribute to an increase in oxidative stress. Different studies have demonstrated that OsaD has a significant impact on human physiology as the changes in oxidative levels of irradiated cells may remain for days after exposure leading to long-term effects [38] in addition to serving as a plasma biomarker [39].

Apart from radiation, the space environment is characterized by reduced gravity that acts as additional stress. Many studies reported that microgravity, real or simulated on ground, affects the physiological level of redox metabolism which in turn causes oxidative stress [40]. Therefore, the effects of space radiation can be exacerbated by a condition of reduced gravity, real or simulated, affecting the OsaD response. The combination of the two stressors can enhance detrimental effects of the space environment, particularly for long-duration missions, where a condition of chronic exposure exists. In this regard, studies derived from cell samples of cosmonauts and space-flown rats have shown an increase in oxidative stress, inflammation, lipid peroxidation, and a decrease of several blood antioxidants [4,41,42]. Alterations in the transcriptional profile of the human immune system were detected in cells such as macrophages and T cells when exposed to an altered gravitational environment [43]. The authors identified a relevant number of oxidative stress-induced gene expression changes between microgravity/hypergravity samples and their respective control cells. Similarly, Overbay et al. [44] found gene expression changes and oxidative damage in the retina of space-flown mice that remained aboard the ISS for 35 days. Persistence of DDR activation, including mitochondrial and oxidative stress, chromosomal aberrations, and telomere elongation has been observed in three unrelated astronauts employed for two 6-month spaceflight missions [4,42]. Loss of function or imbalance in DDR proteins leads to several human diseases including cancer, a risk for spaceflight members exposed to space radiation.

### Repair of DNA Damage Originated in the Space Environment

In space flown cell nuclei, DNA damage was detected using the TdT post-labeling assay and was dependent on the length of the spaceflight [45] and the tracks of double strand breaks were observed by the γ-H2AX foci formation assay [46,47]. The different types of DNA damage induced by space radiation and altered gravity are properly repaired according to their features (Figure 2). For nuclear DNA, different repair pathways are active and well known [3]. Non-homologous end joining (NHEJ) is the predominant pathway to repair radiation-induced DSBs throughout all cell cycle phases (G_1_-S-G_2_) and it is employed by non-proliferating (G_0_) cells. In mammalian cells, NHEJ is sub-divided into CANONICAL NHEJ (c-NHEJ) and alternative NHEJ (alt-NHEJ) that rely on the activity of different specific proteins. Although NHEJ is overall a non-conservative pathway, c-NHEJ generally restores sequence integrity, whereas alt-NHEJ has low fidelity end-joining activity with frequent microhomologies [48]. Homologous recombination (HR) predominates in the mid-S and mid-G2 cell cycle phases of proliferating cells when the sister chromatid is available to repair DSBs in a conservative manner [49]. SSBs originated directly by radiation, or indirectly by reactive oxygen species, are repaired by base excision repair (BER), whose activity uses DNA glycosylases to remove damaged bases producing abasic sites that are filled and repaired by DNA polymerase beta and DNA ligase IIIa (reviewed in [50]). BER also intervenes in repairing oxidative DNA damage together with the mismatch repair (MMR) system [34,51,52,53]. Interestingly, oxidized DNA base damage and BER activation promote the formation of G4 structures that, by regulating gene expression, have a role in transcription, replication and other important biological processes [54].

In the space environment, the occurrence of chromosomal translocations in *Drosophila melanogaster* [55] and larval malformations in *Carausius morosus* [56] were dramatically increased in response to microgravity and radiation exposure. Although it has been hypothesized that these synergistic effects might be caused by an interference of microgravity with DNA repair processes, recent studies on bacteria, yeast cells, and human fibroblasts suggest that a disturbance of cellular repair processes in the microgravity environment might not be a complete explanation for the reported synergism of radiation and microgravity [57,58,59,60]. Alternative explanations for the impact of microgravity should be considered due to changes in signal transduction, metabolic/physiological states, chromatin structure at the cellular level, modification of self-assembly, intercellular communication, cell migration, pattern formation, or differentiation at the tissue and organ level [61].

Mitochondrial DNA (mtDNA) is also damaged by the direct and indirect action of radiation and it undergoes repair, although mtDNA repair mechanisms are less characterized. Evidence has shown that BER activity, carried out by the same proteins active in the nucleus, is also predominant in mitochondria to repair oxidized bases and AP sites [62]. Mammalian mitochondria also contain proteins for nucleotide excision repair (NER), the NHEJ and HR repair pathways, but the repair activities of these pathways are not clearly demonstrated and are still under investigation. The presence of MMR proteins has been reported in mammalian mitochondria, but the activity of this pathway remains elusive [63].

## 4. Genetic and Epigenetic Changes in Space-Flown Human Cells

The direct effects of IR combined with indirect effects originated from the OsaD condition experienced during space missions can alter the genetic landscape. OsaD is a well-known factor contributing to the dysregulation of several cellular processes such as the immune system, cardiovascular system, endothelial system, bone maintenance, and metabolism [28], however the molecular basis of OsaD pathway regulation under space conditions are poorly understood. Human cells exposed to the space environment undergo genetic changes and alterations to epigenetic factors such as chromatin structure, DNA methylation, histone post-translation modifications (PTMs), and non-coding RNAs (ncRNAs) which can have an impact on gene regulation putting the maintenance of the human epigenome at risk. Here we review genomic alterations detected in human space-flown cells, that are summarized in Table 1.

### 4.1. Genetic Polymorphism

Genetic polymorphism in the form of single nucleotide polymorphisms (SNP) in genes of the DDR pathway can affect an individual’s response to genotoxic stress. Genetic changes induced by short-term spaceflight conditions were observed in *Staphylococcus aureus* [64] and other studies [65]. The NASA Twin study revealed specific SNPs to have been affected during spaceflight which has one component of space radiation damage. Specifically, in relationship to Neuro-ocular issues that occur during spaceflight, it has been shown in the clinic that five SNPs found in risk alleles can predict potential onset of ophthalmic issues [66]. From the NASA Twin study, it was found that six out of the nine risk alleles were present [67]. This is one example of the potential impact of space radiation on genetic polymorphisms and the need to consider mitigation against it.

### 4.2. Chromatin Structure

Chromatin is a dynamic structure whose changes are driven by the action of various factors including chromatin remodelers, chromatin modifying/binding enzymes, and non-coding RNAs as recently reviewed [68]. Mechanical forces applied at the cell surface might act at a distance to promote mechanochemical conversion in the nucleus [69], thus it is expected that space microgravity can affect nuclear shape and chromatin organization with important consequences on gene expression regulation. Under conditions of space weightlessness, chromatin structure of human breast carcinoma cells showed larger and more numerous areas of low density compared with 1-g inflight and ground controls [70]. Although the time was relatively short (48 h after launching), it is indicative of structural changes that inevitably are linked to genome organization.

The yield of chromosome aberrations has been known to increase in lymphocytes from astronauts and cosmonauts after long-duration missions of several months in space [71,72,73,74,75]. Although individual radiosensitivity and background chromosome aberration rates are highly variable, an ex vivo dose-response in pre-flight blood samples from astronauts can be used to predict the chromosome aberration rate during the mission [76]. Since chromosome exchanges, especially translocations, are positively correlated with many cancers [77,78], it is a good biomarker to assess cancer risk for the astronauts. Ground-based studies also showed increased chromosome aberrations with simultaneously exposed to radiation and simulated microgravity [79,80,81].

The ends of human chromosomes are capped by telomeres which preserve genome stability. Telomere-length has been studied with lymphocytes from astronauts’ blood. Telomeres were longer during spaceflight irrespective of mission duration and telomere length shortened rapidly upon return to Earth. Overall astronauts had shorter telomeres after spaceflight than they did before [3,4,42]. Although the definitive mechanism and health effects of telomere length dynamics are not yet determined, there are potential risks of aging as well as general health and disease.

### 4.3. DNA Methylation and Histone PTMs

DNA methylation is the major epigenetic mechanism in regulating gene expression but also histone post-translation modifications (PTMs) are involved in the epigenetic control of gene function. Changes in methylation of histone H3 were detected in human blood-derived stem cells (BDSCs) that remained aboard the ISS for 48–72 h under low gravity [82]. In particular, epigenetic modifications at H3K4me3, H3K27me2/3, and H3K9me2/3 residues, that are involved in gene expression regulation and cellular reprogramming, occur in ground-based incubated cells but not in space-flown cells. Histone H3 acetylation was reduced in human T lymphocytes experiencing hypergravity during the launch phase [83]. In the 1-year NASA twin study mission, DNA methylation of blood lymphocytes were examined. Although genome-wide changes were shown to be minimal, DNA methylation changes in immune and oxidative stress–related pathways were observed [3]. In space flown (37 days) mouse retina, a large number of genes were differentially methylated with spaceflight. Particularly profound effects in two important biological processes were found in the ECM/cell junction and for cell proliferation/apoptosis signaling in the retina [84].

### 4.4. Non-Coding RNAs (ncRNAs)

Non-coding RNAs (ncRNAs) are essential factors that contribute to epigenetic maintenance by interacting with histone-modifying complexes as well as act as scaffolds for chromatin-modifying complexes taking part in the DDR pathway [68]. Typically, ncRNAs are small RNAs which can range from 16 nt up to 400 nt. Some ncRNAs will be larger in size. There are a variety of ncRNAs with difference sizes that have been discovered which includes: ribosomal RNA (rRNA) ranging from 120 to 4500 nt, transfer RNA (tRNA) ranging from 76 to 90 nt, small nuclear RNA (snRNA) ranging from 100 to 300 nt, small nucleolar RNA (snoRNA) ranging from 60 to 400, telomerase RNA (TERC), tRNA-Derived Fragments (tRF) ranging from 16 to 28 nt, tRNA halves (tiRNA) ranging from 29 to 50 nt, microRNA (miRNA) ranging from 18 to 25 nt, small interfering RNA (siRNA) ranging from 20 to 25 nt, piwi-interacting RNA (piRNA) ranging from 26 to 32 nt, enhancer RNA (eRNA) ranging from 50 to 2000 nt, long non-coding RNAs (lncRNA) greater than 200 nt, circular RNA (circRNA) ranging from 100 to 10,000 nt, and Y RNA [85]. In general, although many of these ncRNAs are being studied in diseases, there is limited knowledge on how these ncRNAs are impacted in space and more specifically with space radiation.

Among ncRNAs, microRNAs (miRNAs) modulate post-transcriptional gene expression by physically interacting with target mRNAs, repressing their translation and/or inducing their destabilization and decay [86,87]. Among all the ncRNAs, miRNAs are the most studied related to space biology. Many human miRNAs have been found to be dysregulated in human space-flown cells [2,88] and the first evidence that the space environment regulates miRNA expression comes from the study by Hughes-Fulford et al. [89] carried out in human T cells stimulated with mitogens. During the 11 days in space, the expression of miRNAs involved in immune function and the expression of 85 genes associated with T-cell activation were dysregulated. In another short-duration (>1 month) mission onboard the ISS, human fibroblasts did not show alterations in gene and miRNA expression patterns compared to ground controls [90].

Alterations of miRNA expression in the space environment have however been studied together with gene expression analyses to obtain more informative data. Beheshti et al. [91] identified a spaceflight miRNA signature predicted from utilizing transcriptomic data available from NASA’s GeneLab platform. In this work, they analyzed transcriptomic data on multiple tissues from mice flown to the space in multiple missions which included both space shuttle legacy missions and more recent ISS experiments. From this work they predicted a set of 13 miRNAs were involved with space environment damage and increased health risks. In addition, TGFβ was found to be the central target to the miRNAs. In Malkani et al. [9], further experiments were performed to validate the presence of these circulating miRNAs in both rodents and humans exposed to the space environment. They also inhibited the portion of the miRNAs associated with cardiovascular health risks in space and were able to mitigate space radiation damage, indicating that these miRNAs will not only be a good biomarker to assess health risks in space, but also factors that can be developed into potential countermeasures.

Zhang et al. [90] reported that the space environment, experienced for 3 or 14 days, has a little impact on gene expression changes in human normal non-dividing fibroblasts with respect to ground controls. As observed for gene expression, miRNA expression profiling was affected little in human non-proliferating fibroblasts after being cultured aboard the ISS for 3 and 14 days. The only changes were observed at day 3, with let-7a and miR-29, both down-regulated. Interestingly, let-7a also overlaps with the spaceflight miRNA signature identified and validated in Malkani et al. [9].

### 4.5. General Transcriptomic Changes

Gene expression of human cells is affected by the space environment and these observed changes are unique [92]. Transcripts found to be significantly altered belong to many essential biological processes including immune response, endothelial function, cardiovascular function, bone and muscle dynamics, cell growth, cell death, and oxidative stress response. In HUVEC cells cultured for 10 days on the ISS, Versari et al. [93] identified alterations in focal adhesion, over-expression of TXNIP gene, coding for the thioredoxin-interacting protein, and oxidative phosphorylation, that lead to a condition of oxidative stress that promotes DNA damage and inflammation. By transcriptomic analyses carried out in human adult and neonatal cardiovascular progenitors cultured 30 days aboard the ISS, Camberos et al. [94] detected significant alterations in genes belonging to oxidative stress, with the superoxide dismutase 2 (*SOD2*) gene significantly induced. Pathways for cell proliferation and survival, stemness, senescence, and cardiovascular development were also enriched. Differential gene expression has also been observed in the human myelomonocytic cell line U937 during the 19th DLR Parabolic Flight Campaign [95] where the component of cosmic radiation is limited with respect to that of deep space. Chang et al. [96] demonstrated the downregulation of genes involved in the activation of human T cells which is attributable to the typical immunosuppression observed in astronauts returning on Earth. The transcriptional profiling of selected stress response genes in whole blood from six astronauts that remained in a space shuttle for 10–13 days [97], showed significant gene changes in oxidative stress (*GPX1*, coding for glutathione peroxidase), DNA repair (*XRCC1*, *HHR23A*) and chaperones (*HSP27, HSP90AB1*), compared to pre-flight samples. In the study of Grosse et al. [98], the *PRKAA1* gene was significantly upregulated in human endothelial cells experiencing weightlessness from 31 parabolic flights. This may have a protective role as this gene encodes for a protein involved in the antioxidant status of endothelial cells. The *SOD* gene was induced by 4-fold in real space conditions with respect to 1.2-fold under simulated microgravity [99]. Expression changes in genes in pathways for the oxidative stress response, DNA repair pathways, cell cycle, apoptosis, and energy metabolism were detected in space-flown human fibroblasts after the conclusion of a 4 days and 23 h long spaceflight [100]. 

The effects of longer space missions (>4 months) on molecular dysregulation of human cells derives from the results of the NASA Twin Study [3]. In this study, gene expression profiles resulted significant changes in peripheral blood cells of the twin astronaut at the end of the 1 year-long mission onboard the ISS compared to his twin brother on Earth. Many immune-related pathways were significantly changed (i.e., adaptive immune system, innate immune response, and natural killer cell-mediated immunity) in addition to alterations in inflammatory pathways (higher levels of cytokines that remained elevated during the 6-months after return to Earth). Interestingly, ~91% of differentially expressed genes returned to normal ranges within 6 months post-flight. In this study, higher levels of mitochondrial RNA inflight, as compared with pre- and post-flight, emerged from RNA sequencing. By pathway analyses carried out on multi-omics datasets from NASA’s GeneLab platform derived from space-flown human cells, da Silveira et al. [32] demonstrated that mitochondrial dysregulation was the key common pathway being dysregulated in space. Further analysis in this work demonstrated that this mitochondrial dysfunction had a downstream impact on immune suppression and dysfunction, fatty acid metabolism, cell cycle, and oxidative stress. This might indicate that the observations made from other studies described above potentially is driven by the upstream mitochondrial dysfunction reported in this study.

## 5. Genomic Alterations in Human Cells Cultured in Simulated Space Conditions

Due to the limits of performing experiments in true space conditions, many laboratories around the World have carried out experiments by simulating microgravity on Earth. Data on the combination of simulated microgravity and ionizing radiation with human cells are however limited and most studies considered the effects of microgravity alone. Here we make an overview of current data on genetic and epigenetic alterations detected in human cells under the influence of simulated microgravity, alone or combined with IR, by looking at the relationship with the OsaD response. Data discussed throughout the text are summarized in Table 2.

### 5.1. Nucleotide Structure Variation

Genetic variation is usually generated by mutation. The relative contribution of simulated weightlessness to the generation of mutation has been analyzed for the first time in human T-lymphocytes from monozygotic twins [102] by assessing the in vivo mutant frequency at the hypoxanthine-guanine phosphoribosyl transferase (*HPRT*) locus. Although the effects of simulated weightlessness did not affect the mutant frequency, the authors demonstrated the contribution of individual genetics as a substantial factor in determining mutation frequencies. The mutant frequency at *HPRT* locus increased significantly in human PBLs incubated in MMG during the repair time after 1 Gy and 2 Gy of γ- or X-irradiation compared with those cultured in 1 g [103]. *HPRT* mutant frequency also increased significantly up to 4 Gy in γ-irradiated TK6 cells incubated in MMG compared to 1 g [104]. Notably, the *HPRT* mutant frequency was 2–5-fold higher in cosmonauts of different space missions than in healthy unexposed subjects ([105], Table 1).

Another factor contributing to genetic variation is single nucleotide polymorphisms (SNPs). A small number of recent studies evidenced important findings on genetic polymorphism related to a DNA damage response. Niazi et al. [106] studied the association of DNA repair gene polymorphisms with chromosome aberrations in a healthy population with occupational and personal exposure to genotoxic agents including radiation. The results from a total of 153 DNA repair genes tested in 607 individuals showed SNP associations with chromosome aberration frequency within 14 genes having key roles in maintaining genomic stability. All SNPs were located at a site of strong transcription and 10 of the 14 SNPs influenced the expression of the target DNA repair gene. Therefore, it is expected that a relationship between SNPs and gene expression exists. SNPs in DNA repair genes for HR and MMR pathways involved in repairing IR-induced DNA damage are significantly associated with DNA damage in the form of micronuclei formation in patients treated with radioiodine therapy [107]. Correlation analyses of SNPs in the BER pathway with DNA damage, DNA repair, and mRNA expression in healthy donors also supported a relationship between BER gene polymorphisms and individual radiosensitivity [108]. Based on these results it is expected that individual genetic features will affect the DDR following space radiation exposure.

### 5.2. Chromatin and Histone PTMs Changes

Chromatin architecture influences the accessibility to different proteins such as transcription factors and DNA repair proteins and also has consequences for gene expression and cellular signaling pathways. The DNA sequence and local structure surrounding the 8-oxoG lesion, primarily induced by ROS, has an impact on the competence of repair proteins in the BER and NER systems [109]. Changes in chromatin accessibility for the different proteins involved in the DDR pathway can induce differential activity of the OsaD response. Chromatin condensation and margination associated with expression changes for genes in the Bcl2-apoptosis pathway and for proteins in the PI3K/Akt pathway have been observed in human endothelial cells incubated for 72 h in simulated microgravity [110]. Moreover, genes whose expression is dependent on the linker of the nucleoskeleton and cytoskeleton (LINC) complex, connecting chromatin with the nuclear surface, were altered in human breast epithelial cells cultured under simulated microgravity [111].

Histone PMTs are important epigenetic modifications that control chromatin architecture and gene expression. Wang et al. [112] reported that histone H2B acetylation is induced in human embryonic cells exposed in vitro to shear flow, regulating chromatin dynamics. Moreover, several histone PTMs have been altered by oxidative stress [82,113] and thus it is expected that simulated microgravity, which generates oxidative and mechanic stress, can affect histone PTMs. Analysis of histone H3 methylation at promoter regions of selected genes in the neuronal lineage of human mesenchymal stem cells incubated in simulated microgravity displayed a decrease of H3K27me3 [114].

The activity of chromatin remodelers, such as histone acetyltransferase (HAT) p300, is altered by oxidative stress [113]. Alterations in the expression of histone deacetylase (HDAC) genes under microgravity can have consequences at chromatin level. In human T lymphocytes incubated in simulated microgravity, the expression of HDAC1 showed a decrease at 7 days and correlated with the genome-wide DNA hypomethylation [115]. Dysregulation of HDAC1 (downregulation) and HDAC3 (upregulation) genes were detected in human cancer cells incubated in simulated microgravity for 24–72 h [116].

Telomeric DNA and mitochondrial DNA, based on their physical structure, should be more susceptible to oxidative stress and damage [117]. Under simulated microgravity, transcripts for the telomerase reverse transcriptase (Tert) gene, whose protein is responsible for maintaining telomere ends, and RAD50, whose protein plays a critical role in telomere maintenance, were increased in neonatal cardiac progenitors [118].

Recently, chromatin and epitope changes were mapped at a single-cell resolution in bulk blood and sorted cells from two astronauts (Gertz et al., 2020). These multi-omic (100-plex epitope profile, ATAC-seq, and gene expression) data on PBMCs showed changes in blood cell composition and gene expression post-flight, specifically for monocytes and dendritic cell precursors. These were consistent with flight-induced cytokine and immune system stress, followed by skeletal muscle regeneration in response to gravity. These data also provide a framework for high-resolution immune cell function after spaceflight [119].

### 5.3. DNA Methylation Alterations

Alterations in DNA methylation patterns and in histone PTMs can have a strong impact on gene expression modulation. DNA methylation and histone acetylation were found altered in human T lymphocytes incubated in simulated microgravity [115]. In particular, the expression level of *DNMT1*, *DNMT3a*, and *DNMT3b* genes were time-dependent, increasing when cells were incubated for 72 h in simulated microgravity and decreasing when cells were incubated for 7 days compared with 1g-incubated controls. The study by Chowdhury et al. [120] analyzed the correlation of changes in DNA methylation patterns induced by simulated microgravity with gene expression in human lymphoblastoid TK6 cells. A high number of genomic regions showed alterations in methylation (either increase or decrease) for cells incubated for 48 h in simulated microgravity compared with static controls. In particular, hypermethylation was detected in differentially expressed genes that enriched for the p53 pathway, PI3-kinase pathway, and T/B cell pathway activation, whereas hypomethylation was found in differentially expressed genes for the EGF receptor signaling pathway, apoptosis, and FGF signaling. Although the authors did not find a global trend correlating changes in the genome-wide pattern of 5-methylcytosine (5-mC) and 5-hydroxymethylcytosine (5hmC) with gene expression alterations induced by simulated microgravity, they were able to profile the DNA methylation status of individual transcriptionally up- or down-regulated genes. For example, three genes (*PLIN2*, *MAP3K13*, *FBXO1*, all down-regulated) and two genes (*TSPAN5* and *SPG20*, both up-regulated) were associated with loss-of-5mC at their promoter. This indicates that the relationship between transcriptional activity and DNA methylation is furthermore complicated than expected given the traditional theory that decreases in promoter methylation induce gene activity. These differentially expressed genes were implicated in the mechano-stress response, although the expression of *PLIN2* gene, which codifies for the Perilipin-2 protein related to the production of lipid droplets, has been found to be significantly increased in human HepG2 cells experiencing oxidative stress induced by H_2_O_2_ [121]. The downregulation of *PLIN2* observed in Chowdhury et al. [120] could be related to a lower level of oxidative stress induced by microgravity compared to H_2_O_2_ as well as influenced by the different cell type.

### 5.4. Non-Coding RNAs and Gene Expression Changes

Most data on the dysregulation of ncRNAs under simulated microgravity with or without radiation exposure refer to miRNAs. To gain insight on gene expression alterations, many studies have integrated the microRNAome and transcriptome from the same type of cells derived from normal or tumor samples. In human peripheral blood lymphocytes (PBLs) cultured in normal gravity (1g) or modelled microgravity (MMG) the miRNA expression profiles were significantly altered during the repair time after γ-irradiation (2 Gy) [122]. In particular, a number of radiation-responsive miRNAs decreased in MMG compared with 1g conditions (32 vs. 52 miRNAs, respectively) and several miRNAs (let-7i, miR-7, miR-7-1, miR-27a, miR-144, miR-200a, miR-598, and miR-650) were dysregulated by the combined action of radiation and MMG. Significant alterations emerged in the DDR pathway including the p53-pathway. Significantly enriched Gene Ontology (GO) terms included biological categories of “response to DNA damage stimulus”, “DNA damage response”, and “apoptotic mitochondrial changes” that were enriched only in 1g but not in MMG conditions for PBLs. These results are in accordance with the biological response observed in MMG-incubated PBLs in terms of increased apoptosis, delay in DSB repair, and increase in mutant frequency [103,123]. In the study of Fu et al. [124], human lymphoblastoid TK6 cells were irradiated with the same dose (2 Gy) of gamma rays and incubated for 24 h in simulated microgravity. They showed stressor-specific alterations in the number of differentially expressed long-ncRNAs (lncRNAs) that were associated with differentially expressed genes in the immune/inflammatory response and apoptotic process. In contrast, miRNA profiles were affected very little and the discrepancy is probably related to the normal and tumoral status of the analyzed cells, which reflects different genomic responses towards genotoxic agents. For this study, it is important to note that the low LET (i.e., gamma) irradiation was used for this assessment with microgravity. In Malkani et al. [9], it was shown that miRNAs related specifically to high-LET space radiation response behave vastly different compared to low-LET. The differences with low- and high-LET irradiation might be one explanation for the differences observed with the miRNAs and should be kept in mind when designing future experiments to study space radiation effects.

Significant alterations have emerged when looking at the action of only simulated microgravity on miRNA and gene expression profiles in human cells. According to the cell type, the altered miRNA-mRNA pairs enriched similar biological pathways, although the oxidative stress response seems to manifest after an incubation time >24 h in simulated microgravity. In lymphoblastoid TK6 cells incubated for 72 h in simulated microgravity, Mangala et al. [125] identified seven miRNAs whose expression was significantly changed with respect to control cells incubated in 1g. The genes targeted by these miRNAs were mainly enriched for biological categories of the immune response such as the NF-kB pathway, apoptosis, and survival. MiRNAs targeting genes belonging to cell cycle and proliferation, DNA repair, apoptosis, and the Notch signaling pathway were found to be significantly altered in human colorectal cancer cells and lymphoblast leukemic cells incubated for 72 h in simulated microgravity with respect to the counterpart cultured cells in static 1 g [116]. A group of miRNA-mRNA pairs related to immunity, cell proliferation, and apoptosis were also identified in human PBLs incubated for 24 h in MMG [126]. Many of the genes and miRNAs found to be dysregulated are involved in biological processes of the immune/inflammatory response, signal transduction, regulation of response to stress, regulation of programmed cell death/proliferation, and the NF-kB pathway. Notably, gene and miRNA expression changes enriching, among others, pathways of NF-kB, cell proliferation, inflammation were also identified in HUVEC cells incubated in simulated microgravity for a very short time (i.e., 2 h) [127]. When the same cells were cultured in microgravity for a longer period (i.e., 48 h) dysregulated miRNAs targeted apoptotic genes [128]. One-week incubation in microgravity altered the expression of tissue-specific genes and genes related to apoptosis, cell survival, and proliferation in human mesenchymal stem cells [129].

In the genome-wide study carried out on human lymphoblastoid TK6 cells exposed for 48 h to simulated microgravity, Chowdhury et al. [120] evidenced 370 differentially expressed genes mainly involved in the biological categories of “oxidative stress response”, “carbohydrate metabolism”, and “regulation of transcription”. Prostate cancer cells incubated in simulated microgravity for 3–5 days showed alterations in genes for the VEGF, MAPK, and PAM signaling pathways [130].

Even if gene expression changes observed in real microgravity are unique, genes found to be differentially expressed in human cells that are incubated in simulated microgravity are, in many cases, altered also in real microgravity. However, this is not valid for all cell types, as observed in human neural crest stem cells cultured in real or simulated microgravity where dysregulated genes belong to markedly different pathways [131]. Generally, cells exposed both to real or simulated space conditions reported less marked gene expression changes in simulated than in space conditions, as for the superoxide dismutase (SOD) gene which resulted in a 4.1-fold upregulation in space microgravity respect to 1.2-fold in a rotating wall vessel (RWV) [99].

## 6. Genetic and Epigenetic Changes Associated with Radiation Carcinogenesis Risk

Exposure to the space environment is at the root of the onset for different health problems, among which radiation carcinogenesis is one of the greatest. Such risk is due to the high damaging power of cosmic radiation and the individual genetic landscape, which can influence the radiation response. The radiation carcinogenesis risk associated with deep-space exposure depends, among different factors, on the physical characteristics of cosmic rays. GCRs are constituted by high-LET radiation (protons and HZE ions) that induce direct DNA damage in the form of clustered and complex lesions that are difficult to repair [5,132] and cause chromosome aberrations such as intrachromosomal inversions that continue to rise post-flight [3]. Together, the complex DNA lesions and oxidative DNA damage induced by space radiation and the non-targeted effects of ionizing radiation contribute to the carcinogenesis risk. Importantly, genomic changes are related not only to the spaceflight environment but also to individual genetic features that influence the efficiency of DDR pathway. Thanks to genome-wide studies, new data have emerged in the last few years on genetic and epigenetic alterations human cells face when they are exposed to radiation. High variability in the expression of *XRCC1* and *HHR23A* genes, coding for DNA repair proteins, and the *GPX1* gene important for oxidative stress, was observed in whole blood from astronauts [97]. Similarly, the expression of many genes in the DDR pathway were significantly changed in γ-irradiated single donor-derived PBLs incubated in MMG [122], but several of the same genes were unaffected when PBLs were analyzed as pools of different donors [123]. This confirms the existence of an individual DDR to IR as reported in other works. Pariset et al. [133] found that individuals with a low basal level of DNA damage are, on the one hand, less sensitive to side effects of radiotherapy, and on the other, more sensitive and show a higher number of repair foci towards radiation-induced DNA damage. These two characteristics confer a better clinical outcome and a stronger DNA repair capacity. Other studies reported differences at the DNA repair capacity that are linked to an individual genotype at four SNPs of the BER genes *hOGG1*, *APEi*, *XRCC1*, and *LIGASE1* [108]. Despite the limitation of a small sample size, the results demonstrate a correlation between gene polymorphism and DNA damage capacity, discriminating radiosensitive donors from those with a radio-adaptive response. The correlation between individual radiosensitivity and SNPs in the BER genes (*LIG1* and *NEIL1*) or DNA repair genes in the NHEJ pathway (*XRCC5*, *XRCC6* and *XRCC7*) was reported in human peripheral blood cells of a population exposed to chronic low-level radiation [134]. Notably, a genomic destabilization has been observed in mouse embryonic fibroblasts after the repair of radiation-induced DSBs due to replication stress-induced DSB accumulation caused by oxidative DNA damage [135]. Under this condition, cells were more prone to develop single nucleotide variants (SNV), including radiation-associated SNVs. A similar mutagenic effect would have detrimental consequences on human cells exposed to space radiation, where however an increase in mutant frequency at *HPRT* gene has been reported in cosmonauts [105]. It is also therefore expected that genes responsible for the carcinogenic process might undergo mutation under the combined action of radiation and microgravity.

Assessment of cancer risk in space missions has been historically estimated based on the space radiation quality and quantity. An attempt to assess the risk of human radiation-induced leukemia has been made in the study of Cucinotta and Smirnova [136]. The authors developed a dynamic model of radiogenic leukemia by using a radiation dose and dose rate that coincided with the mean dose rates of continuous irradiation of nuclear workers and patients treated with radiotherapy. The modeling results demonstrate the predictive power of the model; however, it might underestimate the potential risk to astronauts when ignoring the effects of microgravity. In specific, immune system aberrations caused by stressors associated with space travel should be included when estimating risk of cancer mortality [137]. It was reported that normal mice demonstrated significantly increased splenic and thymic atrophy, tumor growth, and metastasis during hind-limb unloading (HU) compared with controls [138,139]. Although HU was developed to enable the study of the adverse consequences of spaceflight, HU may not represent a perfect model of microgravity. Therefore, it will be necessary to verify these results in space-based experiments.

One of the most promising avenues of research on longitudinal and continual estimates of risk for hematological malignancies or cardiovascular disease is with measurement of clonal hematopoiesis (CH), which was reported in astronauts for the first time in 2020 [140]. Here, data from the NASA Twins Study [3] showed that mutations in known CH genes were found to decrease during the year-long mission, but then showed a 3–4% increase in the variant allele frequency (VAF) upon return to Earth of certain clones. Of note, both astronauts showed CH, and even though they were twins, they each showed distinct CH VAF dynamics in different genes (DNMT3A vs. TET2). These data show the unique responses to somatic mutations that may be related to spaceflight, even among identical twins, and also provide a new means by which to track the dynamics of mutations’ clone size over time and potential impact on long-term astronaut health.

## 7. Countermeasures Development

The reduction of oxidative stress caused by RNS and ROS has been widely researched as a means to protect critical targets such as DNA, RNA, and other cellular components. A number of dietary supplements aimed at reducing oxidative stress have been examined in animal models for a number of biological endpoints [141,142]. These include supplements such as N-acetyl cysteine (NAC), ascorbic acid (vitamin C), vitamin E succinate, coenzyme Q10, folic acid, glutathione, alpha-lipoic acid, selenomethionine, a soybean-derived protease inhibitor (Bowman–Birk inhibitor, BBI). Treatment with alpha-lipoic acid reduced the impairment of spatial memory retention in C57BL/6J male mice exposed to 56Fe brain irradiation [143]. Vitamin A inhibited the expression of inflammation-related genes and significantly reduced neoplasms in a Sprague–Dawley skin model following 56Fe ion irradiation [144]. A diet of selenomethionine with or without sodium ascorbate, NAC, alpha-lipoic acid, vitamin E, and coenzyme Q10 reversed the decrease of serum or plasma levels of total antioxidants following gamma-ray or 1GeV/n of 56Fe ion exposure [145]. A reduction of 56Fe radiation-induced cataracts measured by lens opacification was demonstrated in CBA/JCrHsd mice fed with a BBI concentrate or an antioxidant combination of selenomethionine, NAC, ascorbic acid, coenzyme Q10, alpha-lipoic acid, and vitamin E succinate [146]. A new examination of dietary supplementation with 25% dried plum by weight protected mice from gamma-ray or sequential irradiation of proton and 56Fe ions by reducing the expression of genes related to bone resorption and reducing cancellous bone loss presumably due to its high antioxidant capacity and high polyphenolic content [147]. In addition, this study was extended to examine the combined effect of reduced gravity through hind-limb unloading with exposure to ionizing radiation. Hind-limb unloaded C57BL/6J mice treated with 137Cs gamma-ray irradiation and fed a diet supplemented with dried plum prevented increases in markers for bone resorption, inflammation, and oxidative stress [148]. Results such as these suggest dietary supplementation may protect against the negative effects of oxidative stress during spaceflight. In addition, physiological countermeasures due to microgravity have long been implemented for pre-, in-, and post-flight space missions [149]. In a recent review of 22 studies, exercise was also shown to have a mitigating effect on 68 of 72 relevant outcomes that decreased DNA-damage, oxidative stress, and inflammation from radiation exposure [150].

While the risk of occurrence is low, radiation exposure from a solar particle event (SPE) may lead to acute radiation syndromes (ARS) and require intervention with medical countermeasures (reviewed by [151]). A potentially lethal exposure in the lowest dose ranges is attributed to hematopoietic failure (H-ARS). An effective strategy to protect against these levels of exposure is primarily provided by physical shielding. However, should a radiation exposure occur, the early symptoms or prodromal stages of H-ARS include nausea, vomiting, anorexia, and diarrhea. Medical countermeasures are available to reduce these symptoms such as 5-HT3 serotonin antagonists to reduce nausea or Imodium^®^ to prevent diarrhea. Currently, three drugs have been developed and have FDA-approval to increase survival in patients exposed to acute myelosuppressive doses of radiation. Filgrastim (Neupogen^®^ from Amgen) and the sustained release version pegfilgastrim or Peg-G-CSF (Neulasta^®^ from Amgen) are made of recombinant granulocyte colony stimulating factor (G-CSF) which successfully reduced neutropenia from SPE-like proton irradiation in a mouse-model [152]. Lastly, Sargramostim (Leukine^®^ from Partner Therapeutics) is a recombinant human granulocyte macrophage colony stimulating factor (rhGM-CSF) used to treat both adult and pediatric patients exposed to myelosuppressive radiation doses [153].

The combined effects of oxidative stress due to microgravity and space radiation exposure have gained multiple experimental considerations for distinct tissue types such as the cardiovascular system [154,155], central nervous system [156], and immune system [157]. Post-flight carotid arterial stiffness has been previously observed in astronauts [158]. While there is limited research in pharmaceutical countermeasures, beneficial effects have been observed in a rat model of radiation-induced heart disease through treatment with pentoxifylline, a phosphodiesterase inhibitor, and α-tocopherol (vitamin E). This treatment improved myocardial fibrosis and left ventricular function that may reduce cardiac injury [159]. Protection against immune system deregulation presents an important challenge to maintain astronauts’ health. Potential countermeasures including pharmaceutical, probiotic, prebiotic, and medical treatment strategies using antibiotics or antivirals are an active and ongoing area of research. For long-duration spaceflights, the successful identification and implementation of novel dietary as well as pharmaceutical countermeasures will aid in lowering the overall risk associated with exposure to the space environment.

## 8. Conclusions

Individual genetic features are strictly linked to the cellular response to radiation and the risk of developing cancer or other diseases. Genetic polymorphisms in DNA repair genes of healthy individuals affect the capacity in repairing DNA damage putting radiosensitive individuals at higher carcinogenesis risk when exposed to space radiation. Additionally, the extent of epigenetic changes drive by space environment should depend on individual genomic features. Therefore, one of the research priorities is to set up the best methodological approach to identify the molecular signatures of those individuals that are more prone to develop radiation sensitivity, before their employment in long-term space missions. Based on the recent important findings derived from multi-omics data from astronauts employed in space missions long-duration mission on ISS, it appears evident that next-generation genomics and integrative analyses are the most useful strategy to assess inter-individual variations in the physiological response to space environment.

## Figures and Tables

**Figure 1 ijms-22-10507-f001:**
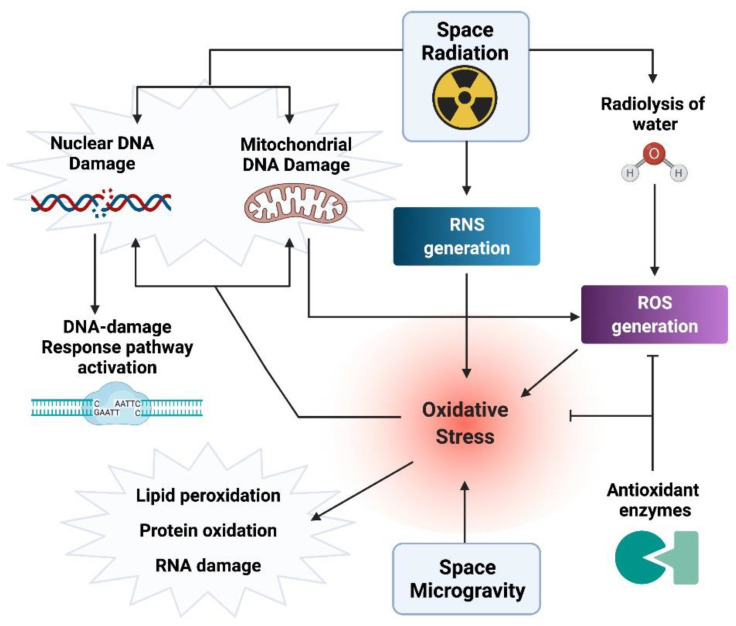
Scheme of the OsaD response in the space environment. The space environment is characterized by a mixture of ionizing radiation of different quality and by reduced gravity (microgravity). Under these conditions, reactive oxygen species (ROS) are generated by radiation-induced water radiolysis and by microgravity causing oxidative stress leading to cellular damage in the form of oxidative DNA/RNA damage, lipid peroxidation, protein oxidation, and impairment of antioxidant activity. Reactive nitrogen species (RNS) induced by radiation also participate in the increase of oxidative stress. Once activated, the OsaD response interfaces with the DNA-Damage Response pathway to counteract the combined effects of radiation and microgravity. “Created with BioRender.com”.

**Figure 2 ijms-22-10507-f002:**
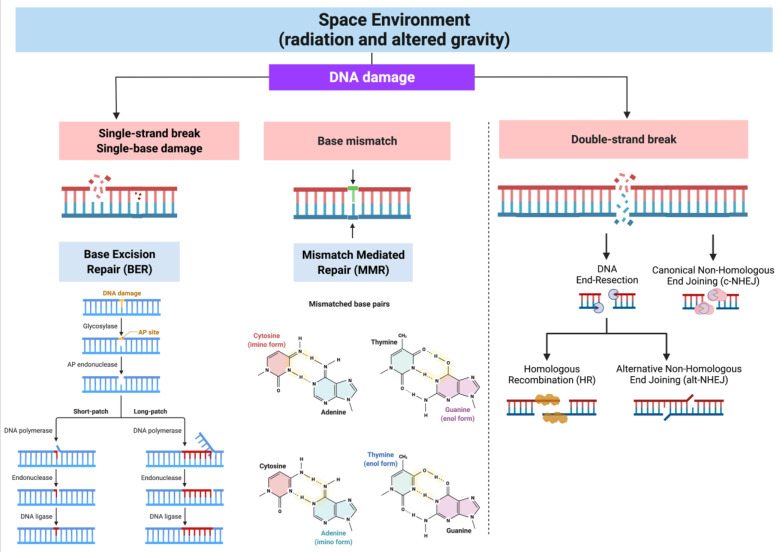
DNA repair pathways associated with DNA damage originated in space environment. The different types of DNA damage induced by ionizing radiation and altered gravity are repaired by components of Base Excision Repair (BER), Mismatch Mediated Repair (MMR), Homologous Recombination (HR) and Non-Homologous End Joining (NHEJ, canonical and alternative) pathways. Oxidative DNA damage in the form of oxidized bases, oxidized pyrimidine derivatives, oxidized base-derived apurinic/apyrimidinic sites are repaired by BER and MMR systems. SSBs and DSBs, originated directly by radiation or indirectly by oxidative DNA damage, are repaired by BER and HR/NHEJ pathways, respectively. “Created with BioRender.com”.

**Table 1 ijms-22-10507-t001:** Genomic alterations in human space-flown cells.

Space Environment	Duration	Type of Cells	Methods	Molecular Alterations	Pathway	References
International Space Station (ISS)	1 year	Human peripheral blood mononuclear cells (PBMCs)	Whole genome sequencing (WGBS)	MiRNA expression Gene expression	Immune/Inflammation-related pathways DNA-damage response	[3]
International Space Station (ISS)	3–14 days	Human normal foreskin fibroblasts (AG1522)	Microarray	MiRNA expression Gene expression	NF-kB pathway Growth-related pathways	[90]
International Space Station (ISS)	1 month	Human normal foreskin fibroblasts (AG1522)	Microarray	Gene expression	DNA-damage response pathway	[46]
Spaceflight (STS-93 mission)	4 days and 23 h	Human normal fibroblasts (WI-38)	cDNA libraries	Gene expression	Apoptosis, senescence	[100]
Soyuz 13S (TMA-09) + ISS	11 days	Human T cells	Microarray	MiRNA expression Gene expression	Immune system	[89]
Space transportation system flight STS-90	6 days	Primary human renal cell cultures	Microarray	Gene expression	Transcription factors	[99]
Soyuz 13S (TMA-09) + ISS	11 days	Human T cells	Microarray	Gene expression	Rel-NF-kB pathway	[96]
SpaceX CRS-11 + ISS	30 days	Human adult and neonatal cardiovascular progenitors	RNA sequencing	MiRNA expression Gene expression	Senescence, stemness, cell proliferation, survival, oxidative stress	[94]
International Space Station (ISS)	5.5 weeks	Human induced pluripotent stem cell-derived cardiomyocytes (hiPSC-CMs)	RNA sequencing	Gene expression	Mitochondrial metabolism; DNA damage and repair	[101]
International Space Station (ISS)	10 days	Human Umbilical Vein Endothelial Cells (HUVECs)	Microarray	Gene expression	Cell adhesion, oxidative phosphorylation, stress responses, cell cycle, apoptosis	[93]
Parabolic flight and suborbital ballistic rocket experiments		Human myelomonocytic cells (U937), Jurkat T cells, primary T lymphocytes			ROS metabolism, antioxidative systems, oxidative stress response	[43,83]
Parabolic flight	22 s	Human endothelial cells (EA.hy926)	Microarray and qRT-PCR	Gene expression		[98]
Space shuttle	10–13 days	Human whole blood	Microarray	Gene expression		[97]
International Space Station (ISS)	48–72 h	Human blood-derived stem cells (BDSCs)		Histone H3 PTMs	Cell differentiation	[82]

VEGF, Vascular Endothelial Growth Factor; MAPK, Mitogen-Activated Protein Kinase; PAM, PI3K/AKT/mTOR. RPM, Random Positioning Machine; RCCS, Rotating Cell Culture System. NGS, Next Generation Sequencing.

**Table 2 ijms-22-10507-t002:** Genomic alterations in human cells incubated in simulated space conditions.

Condition	Duration	Type of Cells	Methods	Molecular Alterations	Signalling Pathway	References
γ-irradiation (0.2–2 Gy) + microgravity (simulated with the RWV)	4 and 24 h	Human peripheral blood lymphocytes (PBLs)	Microarray	MiRNA expression Gene expression	DNA-damage response, p53 pathway	[122]
γ-irradiation (2 Gy) + microgravity (simulated with the RWV)	24 h	Human lymphoblastoid cells (TK6)	Microarray	Gene expression NcRNAs expression MiRNA expression	Apoptosis, immune/inflammatory response, NF-kB pathway, p53 pathway	[124]
γ-irradiation (2 Gy) + microgravity (simulated with the RWV)	24 h	Human PBLs; human lymphoblastoid cells (TK6)	T-cell clonal assay	Mutant frequency	Hypoxanthine-phosphorybosil transferase gene	[103,104]
Microgravity (simulated with the HARV bioreactor)	72 h	Human lymphoblastoid cells (TK6)	Microarray	MiRNA expression	Immune response, NF-kB pathway, apoptosis, survival	[125]
Microgravity (simulated with the RWV)	24 h	Human peripheral blood lymphocytes (PBLs)	Microarray	MiRNA expression Gene expression	Immune system/Inflammation	[126]
Microgravity (simulated with the RWV)	6 days	Primary human renal cell cultures	Microarray	Gene expression	Shear stress response, adhesion, apoptosis, cytoskeleton, differentiation,	[99]
Microgravity (simulated with a Clinostat)	2 h	Human endothelial cells (HUVEC)	Next-Generation Sequencing (NGS)	MiRNA expression Gene expression	NF-kB pathway, inflammation, cell cycle, proliferation, angiogenesis	[127]
Microgravity (simulated with a RPM)	3–5 days	Human prostate cancer cells	qRT-PCR	Gene expression	VEGF, MAPK and PAM signalling pathways	[130]
Microgravity (simulated with a RCCS)	72 h–7 days	Human T-lymphocyte cells	qRT-PCR	Gene expression	DNA methylation, histone acetylation	[115]
Microgravity (simulated with a RCCS)	24–48 h	Human colorectal cancer cells Human lymphoblast leukemia cells	Microarray	Gene expression	Cell cycle regulation, apoptosis, Notch signalling pathway	[116]
Microgravity (simulated with a RCCS)	3 days	Human choroidal vascular endothelial cells	qRT-PCR TEM	Gene expression; chromatin condensation/margination; mitochondria vacuolization	Apoptosis, PI3K/AKT pathway	[110]
Microgravity (simulated with a 2D-clinostat)	48 h	Human HUVEC cells	RNA seq qRT-PCR	MiRNA expression Gene expression	Apoptosis	[128]
Microgravity (simulated with a 3D-clinostat)	2–20 h	Human breast epithelial cells	RNA seq	Gene expression	Cell cycle, cell adhesion, cytoskeleton	[111]
Microgravity (simulated with a RCCS)	48 h	Human lymphoblastoid cells (TK6)	Next-Generation Sequencing (NGS)	DNA methylation; gene expression	Response to oxidative stress, ion transport, DNA-dependent transcription, carbohydrate metabolic processes	[120]
Microgravity (simulated with a RCCS)	1 week	Human mesenchymal stem cells	Microarray	Gene expression	Osteogenic differentiation, cell adhesion/communication, cell cycle, cytoskeleton, immune response	[129]
Microgravity (simulated with a 2D-clinostat)	6–7 days	Human cardiac progenitors	RT-PCR	Gene expression	Telomerase maintenance	[118]
Microgravity (simulated with a 3D Clinostat)	7 days	Human mesenchymal stem cells	Western blotting qRT-PCR	Histone modification	Cytoskeleton, histone modification	[114]

TEM, Transmission Electron Microscopy. RCCS, Rotating Cell Culture System. RWV, Rotating Wall Vessel. NGS, Next Generation Sequencing.

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
