# Peer review of "Genomic Changes Driven by Radiation-Induced DNA Damage and Microgravity in Human Cells"

_ijms, 2021, doi:10.3390/ijms221910507_

Round 1

Reviewer 1 Report

This review addressed a very interesting and important topic. The information gathered in this review is rich and well organized. The authors can provide more graphic description of the topics discussed in this review, for example, the DNA repair pathways associated with DNA damage originated in space environments, including NHEJ (c-NHEJ and alt-NHEJ), HR, BER, etc. Sometimes drawing a scheme figure is very helpful to convey knowledge and understanding of these complicated biological pathways.

Author Response

According to the reviewers suggestion we added a new Figure (Figure 2) describing the DNA repair pathways associated with DNA damage originated in the space environment. 

Reviewer 2 Report

This is an exceptionally good review and balanced assessment of the space environment on genetic and epigenetic alterations of human cells. The article highlights important data collected from studies on human cells and suggests their potential use to estimate the risks for spaceflight members exposed to space radiation. Great Job!

The manuscript is recommended for publication in IJMS.

Author Response

We thank the reviewer for his/her very nice comments.